# Palmoplantar Keratoderma: A Molecular Genetic Analysis of Family Cases

**DOI:** 10.3390/ijms23179576

**Published:** 2022-08-24

**Authors:** Olga Shchagina, Valeriy Fedotov, Tatiana Markova, Olga Shatokhina, Oksana Ryzhkova, Tatiana Fedotova, Aleksander Polyakov

**Affiliations:** 1Research Centre for Medical Genetics, Moskvorechye St., 1, 115522 Moscow, Russia; 2Voronezh Regional Clinical Hospital №1, Moscow Avenue, 151, 394066 Voronezh, Russia

**Keywords:** palmoplantar keratoderma, AQP5, KRT1, KRT9, family cases, DNA diagnostics

## Abstract

Palmoplantar keratoderma is a clinically polymorphic disorder with a heterogeneous etiology characterized by marked hyperkeratotic lesions on the surface of palms and soles. Hereditary forms of palmoplantar keratoderma usually have autosomal dominant inheritance and are caused by mutations in dozens of genes, most of which belong to the keratin family. We carried out clinical and molecular genetic analysis of the affected and healthy members of four families with autosomal dominant palmoplantar keratoderma. In three out of four family cases of autosomal dominant palmoplantar keratoderma, the following molecular genetic causes were established: in two families—previously non-described missense mutations in the AQP5 gene (NM_001651.4): c.369C>G (p.(Asn123Lys)) and c.103T>G (p.(Trp35Gly)); in one family—a described splice site mutation in the KRT9 gene (NM_000226.4): c.31T>G. In one family, the possible cause of palmoplantar keratoderma was detected—a variant in the KRT1 gene (NM_006121.4): c.931G>A (p.(Glu311Lys)).

## 1. Introduction

Palmoplantar keratoderma (PPK) is a generalized term for any constant, marked increase in epidermis thickness on the palms and/or soles. Keratoderma can be an acquired consequence of inflammatory dermatosis, such as psoriasis and eczema, systemic inflammatory and oncological diseases, or regular contact with chemical substances [1].

Hereditary PPK types have numerous genetic variants and are differentiated based on the genealogy analysis, onset age, morphology, and extracutaneous features.

All hereditary PPK forms can be subdivided into two groups: non-syndromic (characterized only by hyperkeratosis), and syndromic, which have a certain array of extracutaneous features caused by the pleiotropic effect of genes. In the case of mutations in the following genes: *KRT6A*, *KRT6B*, *KRT16*, *KRT17*, all or some of the affected family members have pachyonychia and hyperhidrosis; in *KRT14*—dental defects, hyperpigmentation of affected regions, and anhydrous; in *CTSC*—periodontitis and arachnodactyly; in *WNT10A*—palpebral cysts; in *PKP1*—curly hair; in *DSP* and *JUP*—cardiomyopathy and woolly hair, nail abnormalities; mutilating PPK and periorificial keratotic plaques with squamous cell carcinomas on the keratotic areas in Olmsted syndrome (*TRPV3*, *PERP*, *MBTPS2* genes); macerated hyperkeratosis in case of mal de Meleda caused by recessive mutations in the *SLURP1* gene [2,3,4].

Keratins (KRT) are a family of fibrous proteins. They serve as the main component of nails and hair, as well as the outer layer of skin. Two loci of keratin clusters (12q13 and 17q21) contain an array of genes, mutations in which can cause PPK. Locus 12q13 contains the following genes: *KRT1*, *KRT2*, *KRT5*, *KRT6A*, *KRT6B*, *KRT6C*, *AQP5 (Aquaporin 5)*; locus 17q21: *KRT9*, *KRT10*, *KRT16*, *KRT17*, *KRT14*. Thus, if the family size allows the conduct of a linkage analysis, it is possible to exclude approximately half of the candidate genes. This is undoubtedly important, since keratin-encoding genes are highly polymorphic and carry numerous clinically insignificant variants, especially outside the functional domains.

## 2. Results

### 2.1. Family PPK1

Family PPK1 from Voronezh Region and Lipetsk Region had a segregation of palmoplantar keratoderma in four generations. We examined eight patients aged from 6 to 60 years and five healthy blood relatives (Figure 1a). The disorder had manifested in all affected family members during the first year of life. All affected family members had thickened epidermis, as well as hyperhidrosis and recurring dermatophytosis on affected skin zones. The foci of the hyperkeratosis were painless. However, skin cracks and joining infections caused suffering to patients. Examination showed soft yellow diffuse keratoderma on palms and soles, keratolysis and erythematous plaques with a clear demarcation line on the borders of affected zones (Figure 1b).

We carried out linkage analysis using polymorphic markers D17S1814, D17S800, D17S1778 at the 17q21 locus and D12S83, D12S368, D12S85, D12S1661, D12S1586 at the 12q13 locus in the PPK1 family. Linkage with the 17q21 locus was excluded, and linkage with the 12q13 locus was confirmed: LOD (logarithm of the odds) score for the D12S368 marker was 3.69 with θ = 0.00. LOD scores for markers D12S1661 and D12S1586 flanking the keratin cluster on chromosome 12 were less than -2 with θ = 0.00. Thus, the linkage region was limited to 12:48607 kB–12:54146 kB. Whole-exome sequencing (WES) was carried out for Proband IV.1 from this family. The *AQP5* (NM_001651.4): c.369C>G (p.(Asn123Lys)) variant was detected in the linkage region. This variant segregates in conjunction with the disease in the family (LOD 3.7 with θ = 0.00). The c.369C>G nucleotide sequence variant is not present in the gnomAD database and has not been detected in 1335 exomes of Russian patients with various hereditary pathologies according to the “RuExac” database. The pathogenicity evaluation programs show conflicting results: BayesDel_addAF, DANN, EIGEN, LIST-S2, MVP, MutationAssessor, MutationTaster, PrimateAI, and SIFT v3 estimate this variant as benign, while DEOGEN2, FATHMM-MKL, and M-CAP consider it pathogenic. The 123 asparagine is localized on the surface of the aquaporin 5 molecule in the region of its interaction with other aquaporin subunits. A substitution of neutral asparagine for a larger positively charged lysine can lead to disruption of aquaporin interaction with other proteins, but does not affect the main function of the aqueous channel. A c.367A>T (p.(Asn123Tyr) missense mutation in the same codon was described as a cause of PPK in a family from China. A functional analysis of the c.367A>T (p.(Asn123Tyr) variant showed an increased swelling of model mutant cells in comparison to the wild type. This may lead to a hypotonic activation of TRPV4 calcium channels and an increase in concentration of intracellular calcium, which may cause keratinocyte apoptosis to increase, and as a result, hyperkeratosis [5].

Considering all the data, this variant was evaluated as likely pathogenic according to the ACMG criteria and was estimated to be a cause of PPK in this family.

### 2.2. Family PPK2

The PPK2 family of Tatar origin lives in Kazan, Republic of Tatarstan. PPK segregation was traced in four generations (Figure 2a). The disease has autosomal dominant inheritance. We examined three affected family members from three generations. The disease manifested at the age of 1 year. Symptoms: yellow-grey hyperkeratotic lesions with deep fissures on the skin of palms and soles, cutis marmorata on palms and soles, fingertip narrowing, palmoplantar hyperhidrosis, finger flexion contractures, nail deformities, onychodystrophy (Figure 2). Aside from the palmoplantar lesions, all patients have xeroderma, perioral keratosis, fissures in corners of the mouth, sparse hair, hypodontia, dental structure abnormalities (Figure 2b)

Whole-exome sequencing was carried out for Proband IV.1 from the PPK2 family. We detected three heterozygous variants: two previously non-described, *AQP5* (NM_001651.4): c.103T>G (p.(Trp35Gly)) and *KRT10* (NM_000421.5): c.1486G>A (p.(Gly496Ser)), and one previously described as pathogenic: *FLG* (NM_002016.2): c.1501C>T (p.(Arg501Ter)).

We carried out a segregation analysis for all three detected variants in the PPK2 family. It was established that variants *AQP5* (NM_001651.4): c.103T>G and *FLG* (NM_002016.2): c.1501C>T are inherited in conjunction with the disease—these variants were detected in the proband’s affected grandmother II.1 and father III.2 and not detected in the proband’s healthy mother III.1 and paternal halfsib IV.2, while the *KRT10* c.1486G>A variant was inherited from the proband’s healthy mother III.1.

The following variant in exon 1 of the *AQP5* gene (NM_001651.4): c.103T>G (p.(Trp35Gly)) was not registered in gnomAD and was not present in 1335 exomes of Russian patients with various hereditary pathologies. Pathogenicity prediction algorithms show conflicting results: BayesDel_addAF, DEOGEN2, FATHMM-MKL, LIST-S2, M-CAP, MVP, MutationTaster, and SIFT estimate this variant as pathogenic, while DANN, EIGEN, MutationAssessor, and PrimateAI as likely benign. There is one described variant in the same codon of the *AQP5* gene: c.104G>C (p.(Trp35Ser)) in a patient with keratoderma [6].

The following nucleotide sequence variant: *FLG* (NM_002016.2): c.1501C>T (p.(Arg501Ter)) in exon 3 was previously described as pathogenic [7,8] and was registered in gnomAD with a frequency of 0.9386% (in 21 cases in a homozygous state). Mutations in the *FLG (Filaggrin)* gene were described in a heterozygous state in patients with *Ichthyosis vulgaris* (OMIM:146700) and a susceptibility to atopic dermatitis type 2 (OMIM: 605803). Despite the fact that this variant has been described as pathogenic multiple times, its pathogenicity is doubted in some studies [9,10].

Thus, considering all data, the genetic cause of the disease in the LPG2 family is most likely the *AQP5* (NM_001651.4): c.103T>G variant, although the family size does not allow us to use the segregation as proof of pathogenicity and the variant is classified as “variant of uncertain significance” (VUS) according to the pathogenicity criteria. The c.1501C>T variant in the *FLG* gene cannot be a cause of PPK, although it might have a modifying impact on the phenotype, leading to more severe keratoderma, which also affects the skin on the face and body, as well as nails and hair.

Mutations in the *AQP5* gene are described as a cause of palmoplantar keratoderma, Bothnian type (OMIM # 600231). Blaydon et al. described five different missense *AQP5* variants in twelve families with palmoplantar keratoderma from Great Britain, Scotland, and Sweden in 2013 [11], including Swedish families from the coasts of the Bothnian bay of the Baltic Sea, in which this PPK type was first described [12]. Later, mutations in the *AQP5* gene were described as a cause of PPK in families from other countries [6,13].

### 2.3. Family PPK3

Family PPK3 of Tatar origin lives in Agryz, Republic of Tatarstan. PPK was segregated in three generations (Figure 3a). We examined eight affected family members and four healthy relatives. The disease had manifested at the age of 1 to 2 years in all affected family members. Examination revealed soft yellow diffuse painless palmoplantar keratoderma and keratolysis without a clear demarcation line on the borders of affected zones (Figure 3b).

For the PPK3 family, we analyzed the linkage with two main PPK loci (17q21 and 12q13) using the following polymorphic markers: D17S1814, D17S800, D17S1778, D12S83, and D12S368. The disease linkage with the 17q21 locus was proved: maximal LOD score of 3.01 with θ = 0.00 was reached for marker D17S1814. Linkage with the 12q13 locus was excluded. We carried out direct Sanger sequencing for the *KRT9* gene from locus 17q21. As a result, we detected a previously described pathogenic variant in all family members with keratoderma: *KRT9* (NM_000226.4): c.31T>G (p.(Leu11Val)) [14]. This variant was not detected in healthy family members, was not registered in gnomAD, and was not present in 1335 exomes of Russian patients from the “RuExac” database. The c.31T>G variant leads to a p.(Leu11Val) missense mutation, which is located in a non-conservative region of the KRT9 protein, while most *KRT9* variants lead to amino acid substitutions in the conservative sequence of the alpha-helical domain. Fuchs-Telem D. et al. carried out functional analysis of this variant and established that the c.31T>G substitution leads to the formation of a new cryptic donor splice site, which interacts with the acceptor site at cDNA nucleotide 516 and leads to a deletion of 162 amino acids: p.Leu11_Gln172del. Considering all data, this variant is pathogenic.

### 2.4. Family PPK4

Family PPK4 of Russian origin lives in Kargapolsky district of Kurgan Region. The proband—a 3-year-old girl—had hyperkeratosis on her left sole and dry skin on her palms. Palmoplantar keratoderma was also present in the proband’s siblings, mother, grandmother, and maternal relatives (Figure 4a). Keratoderma in all family members was painless and did not present significant discomfort. However, the family consulted a geneticist not because of family hyperkeratosis, but because of proteinuria of unknown genesis—urine protein from 0.38 to 0.91 g/L. A nephrobiopsy showed alterations in the examined glomeruli, which can correspond to an early stage of focal segmental glomerulosclerosis; however, considering the glomerular hypertrophy, a secondary focal segmental glomerulosclerosis as a result of a decrease in the total number of nephrons cannot be excluded. Aside from this, the parents noted striped skin pigmentation, which appeared at the age of 1 year after suntanning, mostly on upper limbs. Examination showed the specific features of the III.3 proband’s phenotype: frontal bossing, bilateral epicanthal folds, large posteriorly rotated auricles, nipple hypertelorism, striped skin hyperpigmentation and depigmentation (Figure 4b).

Whole-exome sequencing was carried out for Proband III.3 from the LPG4 family; as a result, the following variant was detected: *KRT1* (NM_006121.4): c.931G>A (p.(Glu311Lys)). This variant was not registered in gnomAD and was not detected in 1335 exomes of Russian patients with various hereditary pathologies. The p.(Glu311Lys) variant affects a highly conservative intermediate filament region of the KRT1 protein—the helix-termination motif at the end of the 2B segment, which influences the keratin helix formation. Prediction programs (BayesDel_addAF, DANN, DEOGEN2, EIGEN, FATHMM-MKL, LIST-S2, M-CAP, MutationAssessor, MutationTaster and SIFT) evaluated the c.931G>A variant as pathogenic. Another pathogenic variant c.931G>C (p.(Glu311Gln)) was described in the same position in a family case of disseminated hyperkeratosis and palmoplantar keratoderma [15].

Mutations in the *KRT1* gene were described in the case of a rare dominant ichthyosis type—Curth–Macklin form of ichthyosis hystrix (OMIM#146590). This disorder manifests at the age of 1 year and has varying symptoms even in members of one family: from a light form of palmoplantar keratoderma to yellow-brown or gray generalized verrucous, spiky or spine-like hyperkeratotic lesions. It can be diffuse and more prominent on the flexural skin of limbs and the body. The lesions can also look like nevi and be located along Blaschko’s lines [16,17]. Proband III.3 from the PPK4 family had striped skin hyperpigmentation and depigmentation (Figure 4b). However, all described mutations in the *KRT1* gene in the case of Curth–Macklin ichthyosis are nonsense mutations or mutations leading to the formation of a premature stop codon due to a deletion in the gene region coding the keratin tail, whereas our detected variant is a missense substitution. Thus, despite the fact that the presence of this variant in the patient’s genotype fits well as a cause of PPK, its causality in relation to the pigmentation alterations is doubtful. Considering all data, because segregation analysis was unavailable in this case, the *KRT1* (NM_006121.4): c.931G>A (p.(Glu311Lys)) variant was evaluated as likely pathogenic.

Aside from the variant in the *KRT1* gene, the patient had a *C3* (NM_000064.4): c.4148C>A (p.(Thr1383Asn)) variant. The detected nucleotide sequence variant was registered in gnomAD with a frequency of 0.0152% (in Latin Americans—0.04233%). Pathogenicity prediction programs provided conflicting results: M-CAP, MutationAssessor, and SIFT evaluated this variant as pathogenic, whereas BayesDel_addAF, DANN, DEOGEN2, EIGEN, FATHMM-MKL, LIST-S2, MVP, MutationTaster, and PrimateAI consider it as likely neutral. Heterozygous mutations in the *C3* gene were described in patients with susceptibility to atypical hemolytic uremic syndrome (OMIM: 612925) [18]. The nucleotide sequence variant *C3* (NM_000064.4): c.4148C>A (p.(Thr1383Asn)) was described in a heterozygous state as likely pathogenic [19], as a VUS [20], and as benign (according to the results of functional analysis) [21]. We consider this variant to be a VUS, but it can nevertheless explain the proteinuria of unknown genesis and glomerulosclerosis detected in the proband.

Unfortunately, the family refused examination of other affected members and segregation analysis in regard to the following variants: *KRT1* (NM_006121.4): c.931G>A (p.(Glu311Lys)) and *C3* (NM_000064.4): c.4148C>A (p.(Thr1383Asn)), which prevented us from considering the detected nucleotide sequence alterations to be unambiguously causative and fully explain the patient’s phenotype with a combination of the variants.

## 3. Discussion

Palmoplantar keratoderma is a genetically heterogeneous and clinically polymorphic disorder. To date, several dozens of genes have been described, mutations in which cause palmoplantar skin lesions. On the other hand, severe generalized phenotypes involving skin lesions are described in the same genes: various forms of ichthyosis, bullous epidermolysis, pachyonychia congenita, etc. Aside from that, PPK can be an acquired result of an environmental impact or a symptom of an oncological disorder. During the diagnostics and therapy tactics selection, it is of utmost importance to differentiate acquired and hereditary forms of the disorder. In the case of therapy for acquired forms, it is usually sufficient to eliminate the main pathology or environmental factor: chemical agent (arsenic, chlorine solutions) [22], to terminate therapy with prescribed medication (beta glucans, lithium, chemotherapeutic agents) [23], to correct metabolic impairments caused by pregnancy, endocrine pathologies, menopause.

Family history, as well as the availability of the biological material from the maximum possible number of affected and healthy blood relatives, allows detectability of the molecular genetic cause of the disease in the family to be significantly increased and the pathogenicity of the detected variants to be estimated using segregation analysis. In some cases, it allows the PP1 criterion to be regarded not as supporting, but as moderate (PM) and even strong (PS) [24]. In the case of mapping a previously unknown locus of the disease, the linkage is considered proven if the LOD (logarithm of odds) score is greater or equal to 3, which means that the segregation of the disease and the marker in the family being caused by physical linkage is 1000 times more probable than it being a coincidence without a physical linkage. However, the branched pedigree with the number of informative meiosis allowing an LOD score of 3 to be reached is a rare occurrence. In reality, the segregation in the family is just one criterion of the detected variant’s pathogenicity and may be taken into account only in aggregate with other pathogenicity/benignity criteria. Bayrak-Toydemir et al. suggested using a Bayes’ factor (BF) of 100, while noting that even BF = 20 would state that the probability of the co-inheritance of the variant and the disease was 95%, which is acceptable in clinical practice [25]. The impossibility of carrying out thorough phenotyping and genotyping of the affected family members, as is the case with families PPK2 and PPK4, did not allow us to confirm the diagnosis on the molecular genetic level with certainty.

When interpreting the causality and pathogenicity of the variants in the case of PPK, it is essential to remember that in the two main loci of keratin clusters—12q13 and 17q21—multiple genes, mutations in which can cause the disease, are located: in the 12q13 locus—genes *KRT1*, *KRT2*, *KRT5*, *KRT6A*, *KRT6B*, *KRT6C*, *AQP5*; in the 17q21 locus—genes *KRT9*, *KRT10*, *KRT16*, *KRT17*, *KRT14*. Therefore, the target analysis of specific genes from the linkage region is not the best diagnostic approach, especially if the detected variants were not previously described as pathogenic.

It is worth noting that the clinical features in families with variants in the same gene *AQP5* varied in severity: in the PPK1 family with the *AQP5* (NM_001651.4): c.369C>G (p.(Asn123Lys)) variant we observed soft yellow diffuse palmoplantar keratoderma, keratolysis, and erythematous plaques with a clear demarcation line on the borders of the affected zones. A similar clinical picture was described for a c.367A>T, p.Asn123Tyr variant, which affects the same amino acid residue as the variant in the PPK1 family. Cao X et al. described a large Chinese family with a segregation of the disease in three generations. The proband was male with slow progressive thickening of skin on palms and soles, which manifested at the age of 3 years, and moderate hyperhidrosis of palms and soles. Examination showed smooth yellow hyperkeratotic plaques on palms and soles with clear demarcation [5].

However, in the PPK2 family with the *AQP5* (NM_001651.4): c.103T>G (p.(Trp35Gly)) variant, all affected members from three generations had a more severe skin lesion in mutilating keratoderma, the stratum corneum was swollen and had a white spongy appearance. This skin alteration was described in other patients with mutations in the *AQP5* gene only after a prolonged contact with water [11]. Wada et al. described a 32-year-old male with a c.104G>C (Trp35Ser) variant in the *AQP5* gene who had hyperkeratosis and skin desquamation on palms and soles from birth and noted that the skin on his palms shriveled quickly in water. The proband had mild palmar erythema with scales located along or around his palmar and dactylar creases; the dorsal regions of his fingers distal to the metacarpal joints were slightly erythematous with scales; the skin on his metacarpophalangeal joints was reddish. Diffuse erythema and mild hyperkeratosis with thick yellow calluses were observed on the weight-carrying regions of both of his soles. Mild erythema with small scales was detected on the dorsal regions of his soles, around the toenails and along the great toe tendons. Hyperkeratotic lesions shaped as granular papules were located on the sides of his feet, diminishing with distance from the soles [6]. The severe clinical findings in affected PPK2 family members could possibly be a combined effect of variants *AQP5* (NM_001651.4): c.103T>G (p.(Trp35Gly)) and *FLG* (NM_002016.2): c.1501C>T (p.(Arg501Ter)), which co-segregate with hyperkeratosis in the family. Filaggrin is a protein playing a key role in differentiation of epidermal cells and barrier function. It forms during the terminal differentiation of granular epidermal cells, when profilaggrin of keratohyalin granules is proteolytically split into filaggrin molecules, which quickly aggregate with the keratin cytoskeleton, leading to the collapse of granular cells to flat enucleate scales, which form the corneal skin layer. A disruption of corneal scale formation in conjunction with damage to aqueous pores due to the variant in the *AQP5* gene probably leads to a drastic corneal skin layer impairment in three generations of the PPK2 family.

The clinical symptom of hyperkeratosis caused by a variant in the *KRT9* gene in the PPK3 family was diffuse yellow palmoplantar keratoderma, which fully corresponds with the clinical findings in a proband described by D. Fuchs-Telem—a young Ashkenazi Jewish male without a PPK family history. The proband had diffuse yellow palmoplantar keratoderma without any other dermatological findings. The patient did not have hyperhidrosis or hair/nail abnormalities. The cause of the disease was the same as in the LPG3 family: a *KRT9* (NM_000226.4): c.31T>G (p.(Leu11Val)) mutation [14].

During the examination and medical genetic counseling, it is necessary to consider the fact that the patient’s clinical phenotype might be caused by a combination of several pathogenic genetic variants. For example, the severe generalized dermatological phenotype in the PPK2 family was a consequence of a combination of two variants: *AQP5* (NM_001651.4): c.103T>G and *FLG* (NM_002016.2): c.1501C>T, while the skin and kidney abnormalities in the proband from the LPG4 family were caused by independent disorders, most likely connected to the following variants: *KRT1* (NM_006121.4): c.931G>A (p.(Glu311Lys)) and *C3* (NM_000064.4): c.4148C>A (p.(Thr1383Asn)).

## 4. Materials and Methods

The analyzed DNA samples were extracted from peripheral venous blood samples of affected and healthy members of four non-related families with autosomal dominant keratoderma.

Linkage analysis for candidate loci 12q13 and 17q21 was carried out using the following polymorphic microsatellite markers: D17S1814, D17S800, D17S1778, D12S83, D12S368, D12S85, D12S1661, D12S1586, with the amplified fragment length polymorphism (AFLP) method using polyacrylamide gel (PAAG).

The nucleotide sequence of the following genes: *KRT1*, *KRT2*, *KRT5*, *KRT6A*, *KRT6B*, *KRT6C*, *AQP5*, and *KRT9*, was analyzed via direct PCR product sequencing using direct and reverse primers based on fermentative Sanger sequencing. Automated sequencing was carried out on ABIPrism 3500xl (Applied Biosystems, Foster City, CA, USA) according to the manufacturer’s protocol. The design of oligonucleotide primers and probes was conducted in the DNA diagnostic laboratory of the Research Centre for Medical Genetics, and the synthesis in CJSC “Evrogen”, Moscow. The primer sequences were selected according to the GeneBank database.

Whole-exome sequencing (WES) was carried out on a new-generation IlluminaNextSeq 500 sequencer based on paired-end reading (2 × 75 bp). The sample preparation was carried out using selective capturing of DNA fragments from the coding regions of more than 20,000 genes (IlluminaTruSeq ExomeKit). The detected variants were called according to the following website: http://varnomen.hgvs.org/recommendations/DNA (accessed on 14 August 2022).

The sequencing results were processed using a standard automated data analysis algorithm provided by Illumina (https://basespace.illumina.com (accessed on 14 August 2022)). Bioinformatic pathogenicity prediction for previously non-described missense mutations was carried out using VarSome [26]; for the effect on splicing, NetGene2 [27] and Human Splicing Finder [28]. The frequencies of the variants were estimated using the gnomAD genomic data aggregator (https://gnomad.broadinstitute.org/ (accessed on 14 August 2022)) and the exome data of Russian patients from the «RuExac» database [29].

The pathogenicity and causality of the detected nucleotide sequence variants were estimated according to the ACMG recommendations [24]

## 5. Conclusions

Despite the obvious clinical symptoms and (in many cases) a family history of pathology, keratoderma is a hereditary disorder with difficulties in molecular diagnostics. The large number of proteins involved in the formation and functioning of the normal epidermis, as well as the extreme polymorphism of the genes coding these proteins, lead to difficult interpretation of the pathogenicity of the detected genetic variants even after segregation analysis.

In this work, we described two novel heterozygous missense variants in the AQP5 gene: c.369C>G (p.(Asn123Lys); c.367A>T (p.(Asn123Tyr) and one in the KRT1 gene: c.931G>A (p.(Glu311Lys). Evidence of their pathogenicity and causality to PPK were provided. In addition, in one of the Russian families with hyperkeratosis, we identified a previously described “pseudomissense” mutation KRT9: c.31T>G (p.(Leu11Val) which is actually a splicing mutation. Summarizing information about the pathogenicity of previously undescribed variants is really important for the daily work of molecular genetic laboratories engaged in the diagnostic search for the causes of hereditary diseases. 

## Figures and Tables

**Figure 1 ijms-23-09576-f001:**
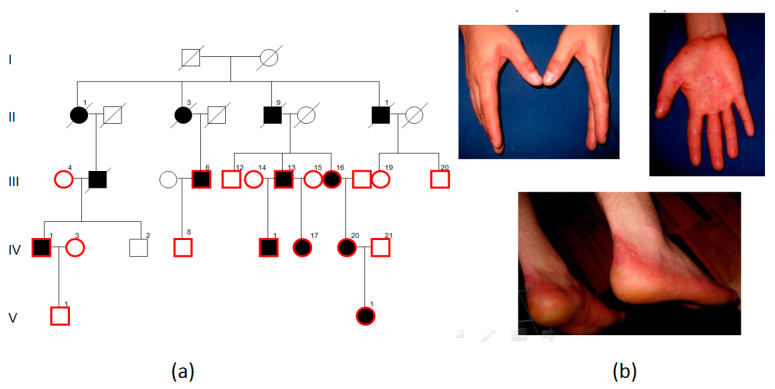
PPK1 pedigree: black—affected family members, white—healthy family members, I–V—generations, 1–21—number of the person in the generation, red contour—symbols corresponding to the examined and genotyped family members (**a**); symptoms of palmoplantar keratoderma in patients: soft yellow diffuse keratoderma on palms and soles, keratolysis and erythematous plaques with a clear demarcation line on the borders of the affected zones of palms and soles (**b**).

**Figure 2 ijms-23-09576-f002:**
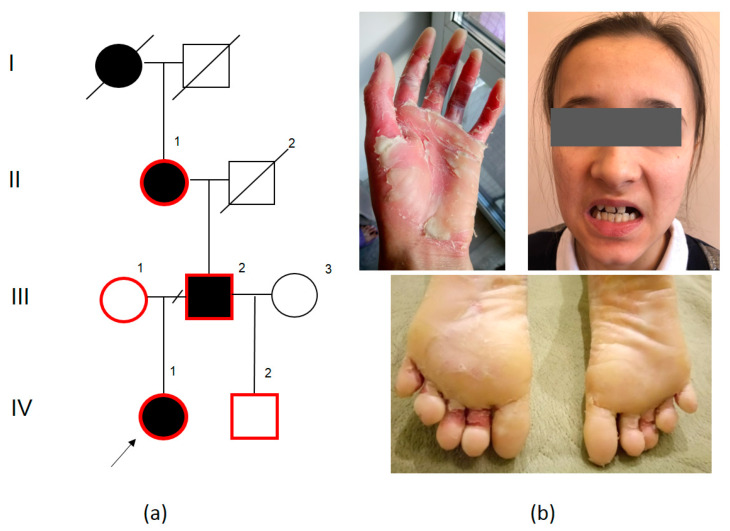
PPK2 pedigree: black—affected family members, white—healthy family members, I–IV—generations, 1–3—number of the person in the generation, red contour—examined and genotyped family members (**a**); symptoms of PPK in patients: diffuse palmoplantar keratoderma, keratolysis, perioral keratosis, fissures in the corners of the mouth, hypodontia (**b**).

**Figure 3 ijms-23-09576-f003:**
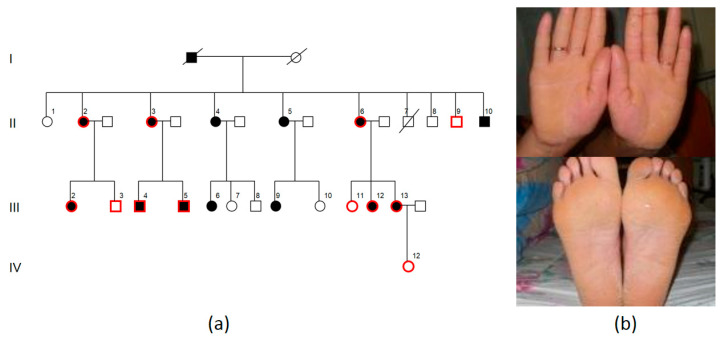
PPK3 pedigree: black—affected family members, white—healthy family members, I–IV—generations, 1–13—number of the person in the generation, red contour—examined and genotyped family members (**a**). PPK symptoms in patients: soft yellow diffuse palmoplantar keratoderma, keratolysis without a clear demarcation line on the borders of affected zones (**b**).

**Figure 4 ijms-23-09576-f004:**
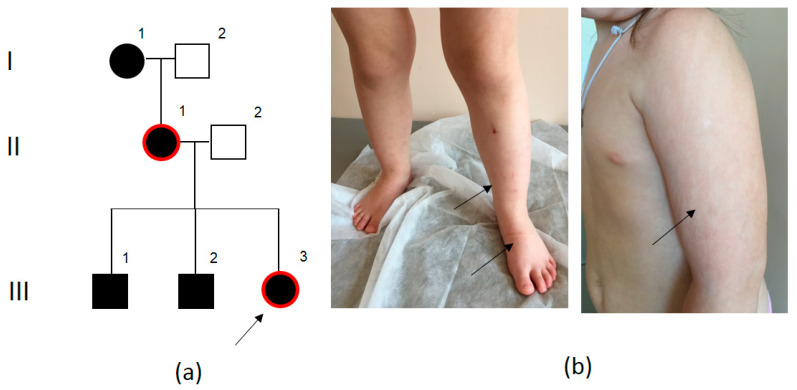
PPK4 pedigree: black—affected family members, white—healthy family members, I–III—generations, 1–3—number of the person in the generation, red contour—examined family members (**a**). Proband: arrows point at striped skin hyperpigmentation and depigmentation (**b**).

## Data Availability

Not applicable.

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
