# Peer review of "Palmoplantar Keratoderma: A Molecular Genetic Analysis of Family Cases"

_ijms, 2022, doi:10.3390/ijms23179576_

Round 1

Reviewer 1 Report

The article « Palmoplantar keratoderma: a molecular genetic analysis of family cases » by Shchagina et al is an interesting description of PPK cases corresponding to 4 families combined with genetic analysis.

In two families the pathogenic gene is considered to be AQP5 whereas in the 2 other families, mutations respectively of KRT9 and KRT1 are reasonably considered to be involved in the pathological presentation.

 I have just a few minor comments or questions :

The authors should detail the abbreviations somewhere, for instance, KRT, VUS…

In the introduction, the authors list various causes of PPK. They omit Olmsted syndrome which is surely a severe but rare disease for which the molecular mechanism has been unraveled (TPRV3 EGFR transactivation) leading to an efficient treatment (EGFR inhibitor erlotinib). EGFR inhibitors show some degree of efficiency in other types of PPK by mechanisms that remain to be investigated.

It would be interesting if the authors could indicate if the PPKs they report are painful. Pain is a very important and deleterious symptom in Olmsted syndrome and Pachyonychia congenitae.

Maybe the authors could prepare a small table or a graph with the 4 mutations they report and mutations of the same codon which were reported in the literature (or if they prefer a review of all mutations in the 3 genes leading to similar pathology).

According to molecular and genetic databases AQP3 is predominantly expressed in salivary glands. Could it explain some aspects of the dental pathology observed in family PPK1 ?

Author Response

We are very grateful to you for the attentive and friendly review of our work. Your questions prompted us to search for additional information. Unfortunately, we could not fulfill all your wishes, but we tried our best.

1) The authors should detail the abbreviations somewhere, for instance, KRT, VUS…

>Thank you for the important remark. All abbreviations are detailed

2) In the introduction, the authors list various causes of PPK. They omit Olmsted syndrome which is surely a severe but rare disease for which the molecular mechanism has been unraveled (TPRV3 EGFR transactivation) leading to an efficient treatment (EGFR inhibitor erlotinib). EGFR inhibitors show some degree of efficiency in other types of PPK by mechanisms that remain to be investigated.

> Oh, this is just a super discussion! Experiments on the functional analysis of missense mutations in the AQP5 gene show a stronger swelling compared to the wild type. Such changes can lead to hypotonic activation of TRPV calcium channels. However, convincing evidence of just such a mechanism of pathogenesis has not yet been obtained. Therefore, indeed, EGFR inhibitors may be effective in the treatment of some forms of PPK. However, given the insufficient scientific and practical basis of these assumptions to date, we do not provide this information in the work (lines 41-43). Information about the Olmsted syndrome was added to the introduction, since this is the only one of the syndromes accompanied by hyperkeratosis that has effective pathogenic therapy.

3)It would be interesting if the authors could indicate if the PPKs they report are painful. Pain is a very important and deleterious symptom in Olmsted syndrome and Pachyonychia congenitae.

> Thanks for the interesting question. We really did not take into account pain as a differential diagnostic sign. Some of our patients complained of soreness only from the appearance of cracks in the skin and the infectious process that has joined. We have added a description of this syndrome in each of the cases (lines 62-63, 165, 203-204).

4)Maybe the authors could prepare a small table or a graph with the 4 mutations they report and mutations of the same codon which were reported in the literature (or if they prefer a review of all mutations in the 3 genes leading to similar pathology).

>Thank you, we really love the tabular format of data representation. Unfortunately, in this case, we could not come up with a table that would not duplicate the information from the text and would not burden the discussion. We analyzed information on all mutations of "our" genes from HGMD - 9 different mutations are presented for the AQP5 gene, KRT1 - 79, in KRT9 - 34 - adding all this information to the article will make it extremely large. In preparing this article, we also processed information about variants of these genes in Russian exomes of patients without complaints of hyperkeratosis. Perhaps we will try to summarize these data in a future review.

5)According to molecular and genetic databases AQP3 is predominantly expressed in salivary glands. Could it explain some aspects of the dental pathology observed in family PPK1 ?

>Thank you for the interesting information for reflection. We do not know whether the pathology of teeth in proband from the PPK1 family is related to the underlying disease. Moreover, problems with teeth of such severity are not reported by other affected family members. In addition, it was in families with a mutation in the AQP5 gene that such a thing was not reported. Although there is information about dental pathology in families with TRPV3 mutations. We do not consider ourselves entitled to explain this feature by the expression of AQP5 in the salivary glands. We have noted these features in this paper, as we believe this information may be useful in the future.

Reviewer 2 Report

 Dear Authors, the text is too long, especially as regard the discussion: the differential diagnosis with the acquired forms of PPK is not necessary. The conclusion, in the other hand, is too short: it must contain the results of the newly identified genetic mutations, which are the only thing that gives importance to the manuscript. the English language needs to be revised 

Author Response

Thank you for the brief but extremely informative review. All comments are taken into account.

Dear Authors, the text is too long, especially as regard the discussion: the differential diagnosis with the acquired forms of PPK is not necessary.

>Thank you for your comment. We have shortened the discussion in the part that concerned acquired forms.

The conclusion, in the other hand, is too short: it must contain the results of the newly identified genetic mutations, which are the only thing that gives importance to the manuscript.

> Thanks for the comment. We have added information about new and previously described genetic variants presented in this paper to the conclusion section (lines 392-399)

the English language needs to be revised 

>We have carried out repeated professional proofreading of our work by a native English speaker.

Round 2

Reviewer 2 Report

the paper can be published.